# Primary complex motor stereotypies are associated with de novo damaging DNA coding mutations that identify *KDM5B* as a risk gene

**Thomas V. Fernandez**[1,2]*, **Zsanett P. Williams**[3], **Tina Kline**[4], **Shreenath Rajendran**[4], **Farhan Augustine**[4], **Nicole Wright**[1], **Catherine A. W. Sullivan**[5], **Emily Olfson**[1], **Sarah B. Abdallah**[1], **Wenzhong Liu**[1], **Ellen J. Hoffman**[1], **Abha R. Gupta**[1,5], **Harvey S. Singer**[4]

1 Yale Child Study Center, Yale University School of Medicine, New Haven, CT, United States America,
2 Department of Psychiatry, Yale University School of Medicine, New Haven, CT, United States America,
3 Department of Psychiatry, Vanderbilt University School of Nursing, Nashville, TN, United States America,
4 Departments of Neurology and Pediatrics, Johns Hopkins University School of Medicine, Baltimore, MD, United States America, 5 Department of Pediatrics, Yale University School of Medicine, New Haven, CT, United States America

* thomas.fernandez@yale.edu

**Data Availability Statement:** The data underlying the results presented in this study have been submitted to the Dryad Digital Repository and

## Abstract

Motor stereotypies are common in children with autism spectrum disorder (ASD), intellectual disability, or sensory deprivation, as well as in typically developing children ("primary" stereotypies, pCMS). The precise pathophysiological mechanism for motor stereotypies is unknown, although genetic etiologies have been suggested. In this study, we perform whole-exome DNA sequencing in 129 parent-child trios with pCMS and 853 control trios (118 cases and 750 controls after quality control). We report an increased rate of de novo predicted-damaging DNA coding variants in pCMS versus controls, identifying *KDM5B* as a high-confidence risk gene and estimating 184 genes conferring risk. Genes harboring de novo damaging variants in pCMS probands show significant overlap with those in Tourette syndrome, ASD, and those in ASD probands with high versus low stereotypy scores. An exploratory analysis of these pCMS gene expression patterns finds clustering within the cortex and striatum during early mid-fetal development. Exploratory gene ontology and network analyses highlight functional convergence in calcium ion transport, demethylation, cell signaling, cell cycle and development. Continued sequencing of pCMS trios will identify additional risk genes and provide greater insights into biological mechanisms of stereotypies across diagnostic boundaries.

## Introduction

Motor stereotypies are rhythmic, repetitive, prolonged, fixed, patterned, non-goal-directed movements that are often bilateral and temporarily stop with distraction. Complex motor stereotypies (CMS) include hand flapping, finger wiggling, head nodding, and rocking; these are

assigned a unique DOI (doi:10.5061/dryad.
rfj6q57d5). The data submission is currently in
"private for peer review" status, so this DOI will not
be live until the manuscript is accepted for
publication. However, a private URL to this data is
provided by Dryad for use during peer review:
https://datadryad.org/stash/share/HU8cwTlay
7QNbwTWYhuuBiZcSeV75dtgmkzsB4B08N0
Clicking this link immediately launches a download
of the data files in the repository.

**Funding:** This work was supported by grants from
the Simons Foundation (sfari.org, SFARI award
#239013, TVF), the Allison Family Foundation
(TVF), Nesbitt-McMaster Foundation (HSS), Klump
Family (HSS), and Graves Family (HSS). The
funders had no role in study design, data collection
and analysis, decision to publish, or preparation of
the manuscript. There was no additional external
funding received for this study.

**Competing interests:** I have read the journal's
policy, and the authors of this manuscript have the
following competing interests: Dr. Fernandez
receives research/grant support from the National
Institutes of Mental Health. Dr. Olfson receives
research support from the National Institutes of
Mental Health, the Alan B. Slifka Foundation
through the Riva Ariella Ritvo endowment, and the
International Obsessive-Compulsive Disorder
Foundation. Dr. Singer serves as a consultant for
Abide Therapeutics, Inc; Cello Health
BioConsulting; ClearView Healthcare Partners; Teva
Pharmaceutical Industries Ltd; and Trinity Partners,
LLC. Dr. Singer receives publishing royalties from
Elsevier and research/grant support from the
Tourette Association of America. Other authors
declare no potential conflicts. This does not alter
our adherence to PLOS ONE policies on sharing
data and materials.

often accompanied by mouth opening, head posturing, jumping, pacing, and occasional vocalizations [1]. Movements occur for up to minutes in duration, multiple times per day, and tend to be exacerbated by excitement, fatigue, stress, boredom, or being engrossed in an activity. CMS are common in children with autism spectrum disorder (ASD), intellectual disability, or sensory deprivation, as well as in typically developing children. A favored classification subdivides by etiology into primary (otherwise typically developing) and secondary categories. In both groups, stereotypies often result in social stigmatization, classroom disruption, and interference with academic activities.

In children with ASD, stereotypic behaviors ("secondary" stereotypies) occur in about 44% of patients and are recognized as a core phenotype of the disorder [2]. The severity and frequency of motor stereotypies is correlated with severity of illness, degree of intellectual disability, and impairments in adaptive functioning and symbolic play [3–9]. They are often associated with self-injurious behaviors [10, 11]. A wide range of medications have been tried for treatment of stereotypies in ASD, but efficacy is inconsistent and inadequate, with potential for long-term side effects [12].

Motor stereotypies also occur in otherwise typically developing children ("primary" stereotypies) [13–22]. Studies comparing primary and secondary stereotypies show that there is considerable similarity in their phenomenology [23–25]. Primary CMS (pCMS) has a typical age of onset before 3 years, and greater than 90% of children continue to experience CMS into adolescence and adulthood [16, 26, 27]. The prevalence of pCMS is estimated to be 3–4% of children in the U.S. [17, 26]. Similar to secondary stereotypies, medications are generally regarded as ineffective for primary CMS [13, 27], but there is evidence to support the benefits of cognitive behavioral therapy [28–30].

The precise pathophysiological mechanism for motor stereotypies remains obscure [31], though investigators have hypothesized abnormalities within cortico-striatal-thalamo-cortical pathways [32–38] and several neurotransmitter systems [33, 39–41]. A recent study reported reduced functional connectivity between prefrontal cortical and striatal regions in pCMS [42]. A genetic etiology for stereotypies has been suggested in primary and secondary categories, although the specific gene(s) contributing to this movement disorder remain unclear. With respect to secondary stereotypies in ASD, family studies have demonstrated that these repetitive behaviors are highly heritable, with a genetic etiology that is likely independent from other core diagnostic features [43]. While there are no studies of recurrence risk or twin concordance reported for pCMS, a positive family history is reported in 25–40%, while remaining cases appear to be sporadic [16, 27, 44].

Considering these findings, we conducted the first pilot genetic study of pCMS in 129 typically developing children and their parents. We hypothesized that pCMS may represent a more genetically homogenous group of individuals versus those with secondary stereotypies, thereby facilitating genetic discovery and insight into the biology of stereotypies more generally [48, 49]. We studied rare de novo, or spontaneous, germline DNA mutations in these individuals. In disorders such as autism, obsessive-compulsive disorder, and Tourette syndrome [45–47], this approach has proven invaluable for identifying genetic variants of large effect, high confidence risk genes, and enriched biological functions. Using whole-exome DNA sequencing, we identified an enrichment of de novo predicted-damaging coding mutations in pCMS and identified one high-confidence risk gene, *Lysine Demethylase 5B* (*KDM5B*) in our cohort. By further analysis of de novo damaging mutations in pCMS, we predict that there are approximately 184 pCMS risk genes and that sequencing more pCMS parent-child trios is a definite path toward discovering these genes. In this pilot study, we see a significant overlap between genes harboring de novo damaging mutations in pCMS and those in ASD as well as Tourette syndrome, a neurodevelopmental movement disorder characterized by motor and

vocal tics. This overlap occurred despite excluding subjects with ASD or tics. Furthermore, owing to the two de novo damaging *KDM5B* mutations in our pCMS cohort, there is significant genetic overlap with ASD probands with highest stereotypy scores, but not those with low scores. Exploratory systems analyses of genes harboring de novo damaging mutations in pCMS show these genes to have peak expression in the cortex and striatum during early mid-fetal development. Finally, exploratory gene ontology analysis highlights functional convergence in calcium ion signaling, demethylation, cell signaling, cell cycle and development.

## Materials and methods

Fig 1 provides an overview of the study methods.

### Subjects and assessment measures

This protocol was approved by the Johns Hopkins Medicine Institutional Review Board. Children with primary complex motor stereotypies (pCMS) were recruited from either the Johns Hopkins Pediatric Neurology Movement Disorder Outpatient Clinic (HSS, Director), or via email (singerlab@jhmi.edu). All participants verbally consented and provided signed parental consent. Using standardized forms via telephone, the study coordinator completed a brief screening general history, obtained baseline data about each child's stereotypies, and completed an Autism Spectrum Screening Questionnaire (ASSQ). The presence of stereotypic movements was confirmed, either via direct observation in clinic or by video review (HSS). If the subject passed the screening assessment, additional data was collected on the child and both parents via RedCap, an electronic web-based application for data capture and online questionnaires. The latter included the Stereotypy Severity Scale (Motor and Impairment scores) and comorbidity measures (Multidimensional Anxiety Scale for Children—MASC; ADHD-Rating Scale IV; Conner's Parent Rating Scale—CPRS; Repetitive Behavior Scale-Revised—RBS-R; Children's Yale-Brown Obsessive-Compulsive Scale—CYBOCS; and Social Responsiveness Scale—SRS) (see S1 File).

For this pilot study, we prioritized the study of "simplex" pCMS (children without known family history of affected first or second-degree relatives) to increase the likelihood of detecting de novo DNA sequence variants. Eligibility required participants to have: (a) confirmed complex motor stereotypies; (b) onset before age 3 years; (c) temporary suspension of movements by an external stimulus or distraction. Exclusion criteria included: (a) a total score >13 on the ASSQ or a prior autism spectrum disorder diagnosis; (b) historical evidence supporting the absence of intellectual disability; (c) seizures or a known neurological disorder; and (d) the presence of motor/vocal tics. The presence of inattentiveness, hyperactivity, or impulsivity (i.e., ADHD symptoms) and/or obsessive-compulsive behaviors were not exclusionary.

### DNA whole-exome sequencing (WES)

DNA was collected from all children meeting eligibility criteria and from their parents, using the Oragene OG-500 collection kit and standard extraction protocols (DNA Genotek, Ottowa, Ontario, Canada). Exome capture and sequencing were performed at the Yale Center for Genome Analysis (YCGA), using the NimbleGen SeqCap EZExomeV2 capture library (Roche NimbleGen, Madison, WI, USA) and the Illumina HiSeq 2500 platform (Illumina, San Diego, CA, USA). WES data from 853 unaffected parent-child trios (2,559 samples total) were obtained from the Simons Simplex Collection via the NIH Data Archive (https://ndar.nih.gov/edit_collection.html?id=2042). These children and their parents have no evidence of autism spectrum or other neurodevelopmental disorders [48]. The same exome capture and sequencing platforms were used for these control samples.

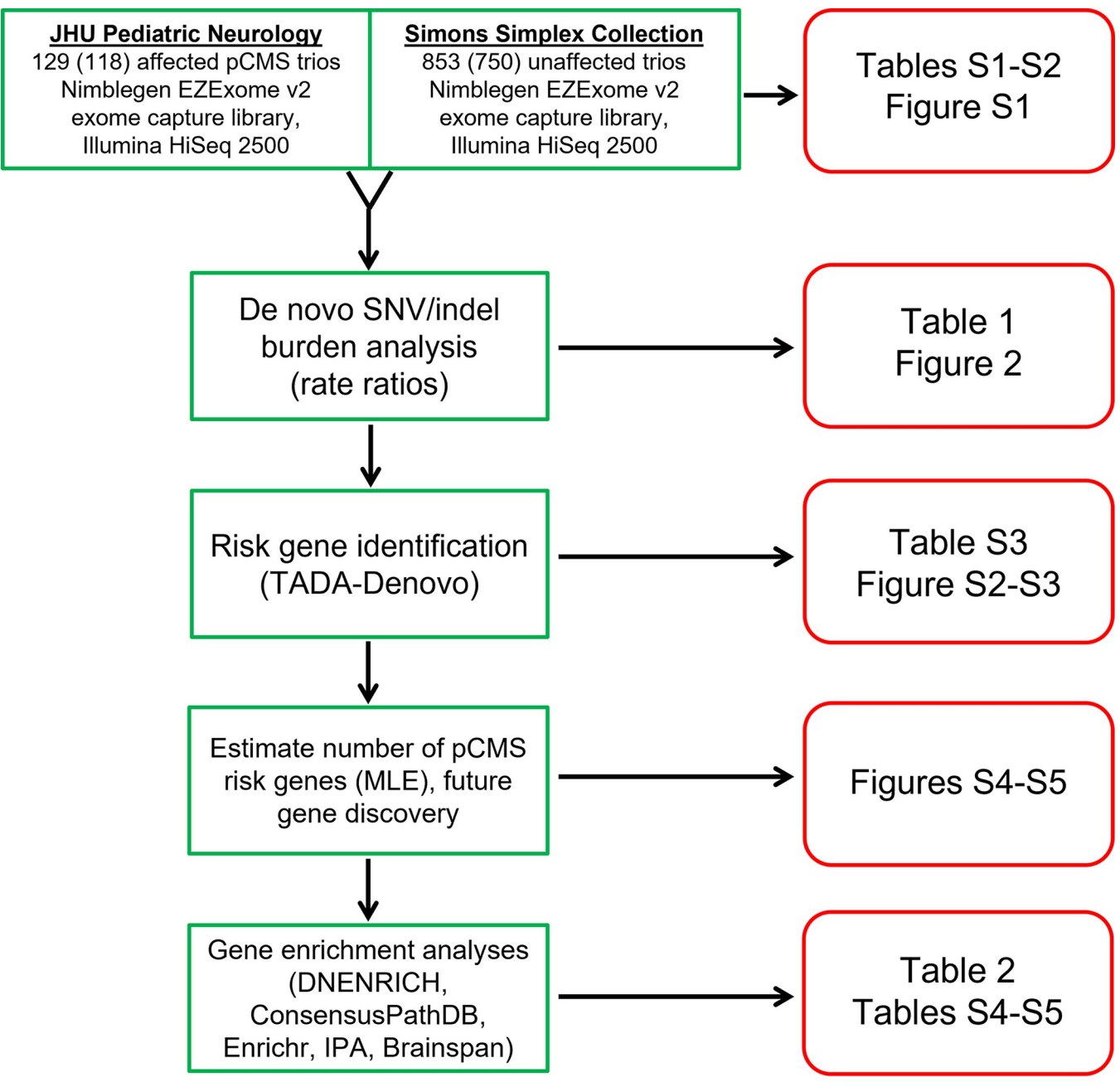

**Fig 1. Overview of variant discovery and data analysis.** We performed whole-exome DNA sequencing of 129 pCMS and 853 control parent-child trios. After quality control, 118 pCMS and 750 control trios remained for subsequent analyses. We performed a burden analysis, comparing the rates of de novo single nucleotide (SNVs) and insertion-deletion (indel) DNA variants between cases and controls. Next, we assessed the statistical significance of gene-level recurrence of de novo damaging variants in our pCMS group, identifying one high-confidence risk gene. Using the maximum likelihood estimation (MLE) method, de novo variant simulations, and TADA, we estimated the number of genes contributing to pCMS risk and used this estimate to predict the number of risk genes that will be discovered as more pCMS trios are sequenced. Finally, exploratory gene enrichment analyses were performed, assessing degree of overlap with gene sets harboring de novo damaging variants in other disorders, gene ontology terms, networks, and expression pattern clustering within certain brain regions across development.

## Sequence alignment, variant calling, and quality control

Alignment and variant calling of the sequencing reads followed the latest Genome Analysis Toolkit (GATK) [49] Best Practices guidelines, as described previously [46]. Variants were annotated using RefSeq hg19 gene definitions using ANNOVAR [50]. Trios were omitted from downstream analyses if (a) genetic markers were not consistent with expected family relationships; (b) an excessive number of de novo variants were observed, or (c) if they were outliers in principal components analysis (see S1 File). De novo variants were called and confirmed as previously described [46] and as detailed in S1 File.

## Mutation rate and gene recurrence

Within each cohort, we calculated the rate of de novo DNA mutations per base pair, using methods previously described [46]. We included only those de novo variants present with a frequency of $<0.001$ (0.1%) in the ExAC v0.3.1 database [51] and compared de novo mutation rates in cases versus controls using a one-tailed rate ratio test (S1 File). Because our cases and controls were sequenced at different times, we took precautions to ensure that batch effects, including differences in sequencing depth and quality, did not influence our comparisons. First, we compared cases and controls that were sequenced on the same sequencing platform and using the same capture library. Second, we considered only "callable" bases, defined as loci with $\geq$ 20x sequencing depth in all family members, with base quality $\geq$ 20, and map quality $\geq$ 30; these thresholds match those required for GATK and de novo variant calling. Third, for each cohort, we summed the "callable" base pairs in every family and used this number as the denominator for de novo rate calculations. In this way, we normalized the de novo rates to guard against any residual differences in sequencing depth or quality, and we compared these normalized rates between cases and controls. This method of comparing different batches of sequencing data has been used in several prior studies [45, 46, 52, 53].

As described in our previous WES studies [45, 46, 52], we used the Transmitted And De novo Association (TADA-Denovo) test as a statistical method for risk gene discovery based on gene-level recurrence of de novo mutations within the classes of variants that we found enriched in pCMS [54, 55]. This test generates random mutational data based on each gene's specified mutation rate to determine null distributions, then calculates a p-value and a false discovery rate (FDR) q-value for each gene using a Bayesian "direct posterior approach." A low q-value represents strong evidence for pCMS association. See S1 File for details.

## Estimating the number of pCMS risk genes

As described previously [46, 52], we used a maximum likelihood estimation (MLE) method [56] to estimate the number of genes contributing risk to pCMS, based on the observed number of de novo damaging variants in our dataset. See S1 File for details of these calculations.

Next, we used previously described methods [46, 52] to predict the likely number of risk genes that will be discovered as additional pCMS parent-child trios are sequenced by WES. These predictions utilize the estimated number of pCMS risk genes along with pCMS de novo mutation rates observed in our study to perform mutation simulations, followed by TADA-Denovo testing (see S1 File).

## Gene set overlap

We used DNENRICH [57] (https://statgen.bitbucket.io/dnenrich/index.html) to test whether genes harboring de novo damaging mutations in our pCMS subjects were significantly enriched among genes harboring de novo damaging mutations in several neuropsychiatric

disorders, including autism (ASD), schizophrenia (SCZ), Tourette's disorder (TD), obsessive-compulsive disorder (OCD), developmental disorders (DD), intellectual disability (ID), and epileptic encephalopathy (EE). Additionally, we were interested in the question of whether our pCMS cohort share genes harboring de novo damaging mutations with ASD probands having high versus low stereotypy scores. To approach this question, we assembled lists of genes harboring de novo damaging mutations in ASD probands from the Simons Simplex Collection (SSC) for whom stereotyped behavior scores (Stereotyped Behavior Score from the RBS-R, Repetitive Behavior Scale-Revised) were available. We looked for overlap between our pCMS cohort and those SSC ASD probands with stereotypy scores in the 90[th] percentile (high stereotypies) and those scoring in the 10[th] percentile (low stereotypies). These gene lists are compiled in S4 Table. Further details about gene list curation and DNENRICH methods can be found in S1 File.

### Exploratory gene ontology, network, and spatiotemporal analyses

To determine whether genes harboring de novo damaging variants in pCMS may perform similar biological functions, we used the list of pCMS genes harboring de novo damaging mutations to identify overlap with gene ontologies using two tools: Enrichr (https://maayanlab.cloud/Enrichr/) [58] and ConsensusPathDB (http://cpdb.molgen.mpg.de/). We identified gene ontology and pathway terms with an enrichment p-value < 0.05. We also used Ingenuity Pathway Analysis (IPA, Ingenuity Systems, http://www.ingenuity.com/) to identify potential gene networks based on this same gene list with the lowest likelihood of interactions due to chance.

Finally, using this same list of genes harboring de novo damaging variants in pCMS, we searched for possible enrichment of gene expression within certain brain regions across multiple developmental time periods, using data from the Brainspan Atlas of the Developing Human Brain [59, 60]. See S1 File.

## Results

We performed WES on 129 pCMS parent-child trios (387 samples total) meeting inclusion criteria. WES data from 853 unaffected control trios, already sequenced from the Simons Simplex Collection, were pooled with our pCMS trios for joint variant calling. After quality control methods, our sample size for a burden analysis was 118 pCMS and 750 unaffected trios (Table 1, Fig 1, S1 Table, S1 Fig).

### Increased burden of de novo damaging variants in pCMS

Based on work in other neurodevelopmental disorders, we expected to find an enrichment of de novo likely gene disrupting (LGD) variants (stop codon, frameshift, or canonical splice-site variants) in pCMS probands versus controls. We found a statistically significant increased rate of de novo LGD variants in pCMS cases, confirming our hypothesis (rate ratio [RR] 1.95, 95% Confidence Interval [CI] 1.04–3.50, p = 0.04). Furthermore, de novo variants predicted to be damaging (LGD plus missense variants with Polyphen2-HDIV score <0.957 and ≥0.453) were also over-represented in pCMS probands (RR 1.37, CI 1.05–1.76, p = 0.03). We did not detect a difference in mutation rates for de novo synonymous variants, or when all de novo variants (coding +/- non-coding) were considered together (Table 1, Fig 2, S2 Table).

### *KDM5B* is a high-confidence candidate risk gene in pCMS

Having established a higher rate of de novo damaging variants in pCMS probands, we next asked whether these variants cluster within specific genes. We identified one gene with more

**Table 1. Distribution of de novo variants in pCMS cases and controls.**

| De novo variant type[a] | Variant counts | | Mutation rate (x10⁻⁸) per bp (95% CI)[j] | | Estimated coding variants per individual (95% CI)[k] | | Rate ratio (95% CI) | p-value[l] |
|---|---|---|---|---|---|---|---|---|
| | pCMS (N = 118) | Control (N = 750) | pCMS (N = 118) | Control (N = 750) | pCMS (N = 118) | Control (N = 750) | | |
| **All**[b] | 134 | 666 | 1.91 (1.60–2.27) | 1.68 (1.56–1.82) | 1.29 (1.08–1.54) | 1.14 (1.05–1.23) | 1.14 (0.97–1.33) | 0.10 |
| **Coding**[c] | 128 | 628 | 2.02 (1.68–2.40) | 1.74 (1.61–1.88) | 1.37 (1.14–1.62) | 1.18 (1.09–1.27) | 1.16 (0.98–1.36) | 0.07 |
| **Synonymous SNV** | 30 | 171 | 0.47 (0.32–0.68) | 0.47 (0.41–0.55) | 0.32 (0.22–0.46) | 0.32 (0.28–0.37) | 1.00 (0.70–1.40) | 0.50 |
| **Nonsynonymous**[d] | 96 | 446 | 1.51 (1.23–1.85) | 1.24 (1.12–1.36) | 1.02 (0.83–1.25) | 0.84 (0.76–0.92) | **1.23 (1.01–1.48)** | **0.04** |
| **All Missense (Mis)** | 84 | 411 | 1.32 (1.06–1.64) | 1.14 (1.03–1.25) | 0.89 (0.72–1.11) | 0.77 (0.70–0.85) | 1.16 (0.95–1.42) | 0.12 |
| **Mis-D**[e] | 42 | 190 | 0.66 (0.48–0.90) | 0.53 (0.45–0.61) | 0.45 (0.32–0.61) | 0.36 (0.30–0.41) | 1.26 (0.93–1.68) | 0.11 |
| **Mis-P**[f] | 15 | 79 | 0.24 (0.13–0.39) | 0.22 (0.17–0.27) | 0.16 (0.088–0.26) | 0.15 (0.12–0.18) | 1.08 (0.64–1.75) | 0.43 |
| **Mis-B**[g] | 25 | 137 | 0.39 (0.26–0.58) | 0.38 (0.32–0.45) | 0.26 (0.18–0.39) | 0.26 (0.22–0.30) | 1.04 (0.70–1.50) | 0.46 |
| **Likely Gene Disrupting (LGD)**[h] | 12 | 35 | 0.19 (0.098–0.33) | 0.097 (0.068–0.13) | 0.13 (0.066–0.22) | 0.066 (0.046–0.088) | **1.95 (1.04–3.50)** | **0.04** |
| **Damaging (LGD + Mis-D)** | 54 | 225 | 0.85 (0.64–1.11) | 0.62 (0.54–0.71) | 0.58 (0.43–0.75) | 0.42 (0.37–0.48) | **1.37 (1.05–1.76)** | **0.03** |
| **LGD SNV** | 10 | 19 | 0.16 (0.076–0.29) | 0.053 (0.032–0.082) | 0.11 (0.051–0.20) | 0.036 (0.022–0.055) | **3.00 (1.43–6.03)** | **0.007** |
| **LGD Stopgain** | 6 | 16 | 0.095 (0.035–0.21) | 0.044 (0.025–0.072) | 0.064 (0.024–0.14) | 0.030 (0.030–0.049) | 2.13 (0.82–5.02) | 0.10 |
| **LGD Splice** | 4 | 3 | 0.063 (0.017–0.16) | 0.0083 (0.0017–0.024) | 0.043 (0.011–0.11) | 0.0056 (0.0012–0.016) | **7.59 (1.66–38.5)** | **0.01** |
| **LGD frameshift indel** | 2 | 16 | 0.032 (0.0038–0.11) | 0.044 (0.025–0.072) | 0.022 (0.0026–0.074) | 0.030 (0.017–0.049) | 0.71 (0.12–2.56) | 0.77 |
| **Nonframeshift indel** | 1 | 3 | 0.016 (0.0004–0.088) | 0.0083 (0.0017–0.024) | 0.011 (0.00027–0.060) | 0.0056 (0.0012–0.016) | 1.90 (0.07–17.2) | 0.47 |
| **Unknown**[i] | 1 | 8 | 0.016 (0.0004–0.088) | 0.022 (0.010–0.041) | 0.011 (0.00027–0.060) | 0.015 (0.0068–0.028) | 0.71 (0.03–4.28) | 0.77 |

[a]Variants were annotated with Annovar, using RefSeq hg19 gene definitions.

[b]"All" includes coding and non-coding variants.

[c]"Coding" variants include synonymous, nonsynonymous, nonframeshift, and those annotated as "unknown" by Annovar.

[d]"Nonsynonymous" variants include all missense and LGD variants.

[e]"Mis-D" are "probably damaging" missense variants with a Polyphen2 (HDIV) score ≥0.957. [f]Mis-P are "possibly damaging" missense variants with a Polyphen2 (HDIV) score <0.957 and ≥0.453. [g]Mis-B are "benign" missense variants with a Polyphen2 (HDIV) score <0.453. Two pCMS missense variants and five control missense variants had no prediction by Polyphen2 but were included in the "All Missense (Mis)" variant type.

[h]LGD variants are those altering a stop codon, canonical splice site, and frameshift indels. [i]"Unknown" variants are not included in the synonymous or nonsynonymous counts.

[j]De novo mutation rates were calculated as the number of variants divided by the number of haploid "callable" bases (see Methods).

[k]The estimated number of de novo mutations per individual was calculated by multiplying the mutation rate by the size of the RefSeq hg19 coding exome (33,828,798 bp).

[l]Rates were compared using a one-sided rate ratio test. Rate ratios, 95% CI, and p-values that are statistically significant (p<0.05) are underlined and in bold. A rate ratio greater than one indicates a higher rate in pCMS versus controls. Also see Fig 2. Variants are listed in S2 Table.

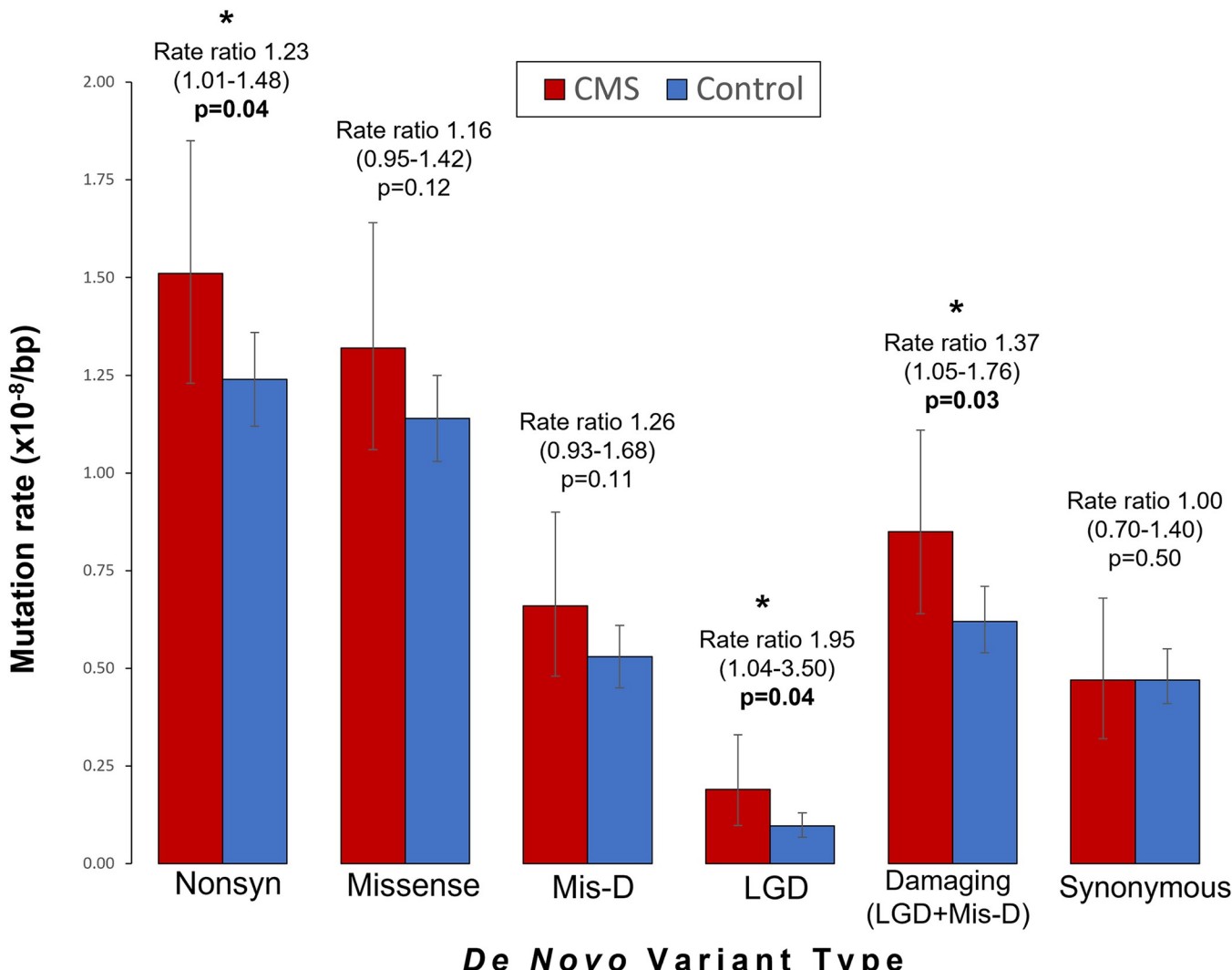

**Fig 2. Rates of de novo variants in pCMS cases versus controls.** Bar chart comparing the rates of de novo variant classes between pCMS cases (red) and controls (blue). Comparisons are between per base pair (bp) mutation rates, using a one-tailed rate ratio test. Statistically significant comparisons (p<0.05) are marked with asterisks. Error bars show 95% confidence intervals.

than one predicted damaging de novo variant in unrelated probands: *KDM5B* (*Lysine Demethylase 5B)* harbored two different LGD (stopgain) de novo variants in pCMS probands 1029–03 and 1050–03. Using TADA-Denovo [54] and previously established false discovery rate (FDR) thresholds, we found that *KDM5B* meets statistical criteria for a high-confidence risk gene (q<0.1) in pCMS (S3 Table).

### Approximately 184 genes contribute to pCMS risk

Based on the number of observed de novo damaging mutations in pCMS, the MLE method estimated the most likely number of pCMS risk genes to be 184 (S2 Fig). Next, we used this estimate along with de novo mutation rates observed in pCMS trios to predict the likely number of these 184 risk genes that will be discovered in larger pCMS cohorts. Based on these simulations, WES of 500 trios should find 16 probable and 7 high-confidence risk genes; 1000 trios should find 51 probable and 26 high-confidence risk genes (S3 Fig).

**Table 2. 52 Genes harboring de novo damaging variants in pCMS and not in controls.**

| | | | | | | |
|---|---|---|---|---|---|---|
| ACACB | BBX | DESI1 | GUCA1B | MRAS | SLC7A7 | ZNF195 |
| ADGRF2 | C16orf87 | DLG5 | HERC1 | NAV3 | TET3 | ZNF461 |
| ADRBK1 | CCDC25 | DNAH6 | ITSN1 | PHIP | TNFRSF10B | ZNF74 |
| AGO4 | CENPP | DPPA5 | KDM3B | PPP1R14C | TRIM55 | ZNF862 |
| ALG8 | COG8 | FAM65B | KDM5B | PRRG4 | TRPM1 | |
| ANKRD39 | COL21A1 | FEZ2 | KDM5B | RAB11FIP3 | TRPV4 | |
| ARVCF | CORO6 | GGCX | LY9 | RHAG | UVSSA | |
| ATP2B2 | DDR1 | GLYR1 | MASP2 | RRBP1 | WNT5A | |

## pCMS gene enrichment in Tourette's disorder, ASD, and in ASD with high versus low stereotypy scores

Using DNENRICH [57], we found significant overlap between genes harboring de novo damaging variants in pCMS (52 genes after excluding two genes with de novo damaging variants in controls, Table 2, S2 Table) and several gene sets curated from the literature (S4 Table). In particular, our pCMS cohort genes show significant gene overlap with autism probands with high stereotypy scores (5.8x enrichment, p = 0.047), Tourette's disorder (4.5x enrichment, p = 0.019), autism spectrum disorder (2.2x enrichment, p = 0.0055–0.0069). There was no significant overlap with OCD, schizophrenia, intellectual disability, developmental disorders, or epileptic encephalopathy (S4 Table).

## Exploratory gene ontology, network, and spatiotemporal analyses

Using this same list of 52 genes harboring de novo damaging variants in pCMS (Table 2, S2 Table), we performed exploratory analyses to identify enrichment in biological, cellular, and molecular gene ontology terms. Using two enrichment tools, we identified significant enrichment for calcium ion transport and demethylation (adjusted p-value < 0.05 in either tool). By relaxing the statistical threshold to an unadjusted p < 0.05, we identified enrichment for these same gene ontology terms in results from both tools (S5 Table). Finally, we performed an exploratory gene network analysis of these 52 genes using IPA and identified the potential importance of these genes in cell signaling, cellular assembly and organization (S5 Table). Finally, mapping our pCMS de novo damaging variant genes onto the Brainspan Atlas of the Developing Human Brain gene expression data, we see nominal enrichment of gene expression in early mid-fetal cortex and striatum, with a trend toward enrichment in early fetal hippocampus, late mid-fetal cerebellum, and young childhood cerebellum (S5 Table).

## Discussion

Like prior studies of ASD, Tourette's disorder, and OCD, the current study demonstrates that the identification of de novo DNA coding variants will identify risk genes and provide a reliable entry-point into understanding the biology of stereotypies. We are studying otherwise typically developing children with stereotypies (primary CMS), as this may represent a more genetically homogenous group of individuals versus those with secondary stereotypies, thereby facilitating genetic discovery and insight into the biology of stereotypies more generally [61, 62]. Despite our small cohort size, we identified two de novo nonsense mutations in *KDM5B* in unrelated probands, and we show that finding two such independent mutations in our cohort is highly unlikely to be a chance occurrence.

*KDM5B* is a lysine-specific demethylase that removes methyl groups from tri-, di- and monomethylated lysine 4 on histone 3. *KDM5B* acts a transcriptional repressor and has

primarily been implicated in the pathogenesis of cancer [63]. More recently, this gene has also been implicated in congenital heart disease risk, embryonic development, DNA repair, adult cognitive function, and muscle strength [64–69]. *KDM5B* has been identified as high-confidence risk gene in ASD via detection of heterozygous de novo damaging variants in WES studies [47, 55] and for developmental disorders more broadly [70]. Individuals reported in the literature with intellectual disability/developmental delay harboring *KDM5B* mutations often show an autosomal recessive inheritance pattern, including inherited homozygous or compound heterozygous mutations in this gene [71, 72], while heterozygous mutations occur more frequently in probands from the Deciphering Developmental Disorders Study [73]. A recent study by Chen et al. [68] identified *KDM5B* as one of eight genes associated with adult cognitive function through rare protein-truncating and damaging missense variants. Consistent with prior reports, they identified a gene dose effect, whereby individuals with rare heterozygous protein-truncating variants showed higher adult cognitive function measurements compared to those with homozygous damaging mutations [68]. Both this study and another recent report by Huang et al. [69] found a significant association between rare variants in *KDM5B* and hand grip strength, a phenotype related to muscle function, and one of several additional phenotypes found to be associated with *KDM5B* [68]. Interestingly, *KDM5B* has a relatively high rate of protein truncating variants, with a rate of approximately 1 in 1,900 subjects in the UK Biobank sample. This is in contrast to most other genes linked to neurodevelopmental phenotypes. Considering the substantial pleiotropic and gene dosage effects reported in several studies to date, it is interesting to see how different mutations and inheritance patterns in this gene can lead to a spectrum of phenotypic outcomes, including ASD, ID/DD, congenital heart disease, adult cognitive function, muscle strength, and now pCMS in childhood. Our team subsequently interviewed families harboring *KDM5B* mutations in our pCMS child probands and confirmed that there was no evidence for ASD, ID, or congenital heart disease.

Expression of *KDM5B* is normally restricted to the brain and the testis [74]. Within the brain, high expression levels of *KDM5B* are seen in the cerebellum (S4 Fig), and expression across all brain regions is highest prenatally (S5 Fig). Consistent with this data, a recent MRI study from our group found volumetric differences in the cerebellum of children with pCMS versus controls, and these changes correlated with Stereotypy Severity Scores [75]. Similarly, cerebellar volume was correlated with stereotyped activity in a deer mouse animal model with repetitive behaviors [75]. The identification of this risk gene in pCMS suggests that chromatin (dys)regulation of *KDM5B* target genes may be one contributing mechanism underlying stereotypies across diagnostic boundaries. Further studies are warranted to determine the downstream effects of these mutations in the developing brain. These studies are underway in our laboratory.

It is interesting that we find significant overlap between genes harboring de novo damaging mutations in pCMS and those reported in a recent study of Tourette syndrome (S4 Table; 4.5x enrichment, p = 0.019). While we have reported approximately 25% of pCMS patients have co-existing tics [14], we find this overlap with Tourette despite excluding pCMS subjects with co-existing motor or vocal tics (see Methods). Enriched expression of pCMS genes in the cortex and striatum (S5 Table) is also consistent with widely believed involvement of these regions in Tourette syndrome. While OCD was not exclusionary in our pCMS study, we saw no significant gene overlap with OCD. Similarly, we found no significant overlap with SCZ, ID, DD, or EE. We did, however, find significant overlap between pCMS and ASD risk genes (2.2x enrichment, p = 0.006–0.007), despite no evidence of ASD in our subjects.

With regard to stereotypies in ASD, we curated lists of genes harboring de novo damaging mutations in SSC probands with the highest (90th percentile) and lowest (10th percentile)

stereotypies, measured by Stereotyped Behavior Scores (SBS) from the RBS-R. *KDM5B* mutations were found only in SSC probands with high stereotypy scores, yielding 5.8-fold enrichment over expectation (p = 0.047) when compared against our pCMS genes (S4 Table). To further examine the relation between de novo *KDM5B* mutations stereotypies in SSC ASD probands, we compared SBS scores in four probands with *KDM5B* mutations versus 364 age-matched patients without (S6 Fig). Scores were higher in mutation carriers, but this did not reach statistical significance (p = 0.076), likely due to the low number of mutation carriers in this cohort.

In summary, we report an increased burden of de novo damaging heterozygous DNA coding variants in primary complex motor stereotypies. We identified one high-confidence risk gene for pCMS in our pilot cohort and estimate that there are 184 genes conferring risk for this phenotype. Whole-exome sequencing in parent-child pCMS trios provides a reliable way to make progress in gene discovery. Our exploratory analyses of genes harboring de novo damaging mutations in pCMS highlight several gene ontology terms (comprising biological processes, molecular functions, and cellular components), as well as brain regions and developmental time periods. These preliminary findings provide insights into possible etiologies of stereotypies, and this knowledge is a prerequisite for developing new treatments. Further sequencing and mechanistic studies are warranted to understand this phenotype, which has relevance across diagnostic boundaries.

## Supporting information

**S1 Fig. PCA scree and individual plots.** Scree plots following Principal Components Analysis (PCA), showing (A) the percentage of variance captured by each of the first 32 principal components, and (B) the cumulative percentage of variance captured by these same components in the exome metrics data from cases and controls. The "elbow" of the scree plot is visualized to be around the 5th principal component. This was confirmed by the Factominer R code function "estim_ncp()". The first 5 PCs capture over 80% of the variance, and this number of PCs was used to determine PCA outliers during quality control (see S1 Table and S1 File). (C) Individual plots for the first two principal components, based on PCA of exome sequencing quality metrics. pCMS cases are plotted in red, and controls in blue. The first two PCs together capture 56.3% of the variance. R code to generate this data and figure are in S1 File, and individual PC factor values are in S1 Table. This figure includes PCA outliers (>3 standard deviations from the mean in PCs 1–5), which were removed during quality control, prior to further analysis of case-control data.
(TIF)

**S2 Fig. Maximum Likelihood Estimate (MLE) of number of pCMS risk genes.** For each number of possible risk genes between 1–2,500, we conducted 50,000 simulations to determine the number of risk genes that yielded the closest agreement between our observed and simulated data. This MLE method yields an estimate of 184 pCMS risk genes (red vertical line). See S1 File.
(TIF)

**S3 Fig. Gene discovery by number of trios sequenced.** Using our estimate of 184 risk genes (based on the MLE method–see Main Text and S1 File), we estimated the number of probable (FDR q<0.3) and high-confidence (FDR q<0.1) risk genes that will be discovered as more pCMS trios are sequenced. We performed 10,000 simulations at each cohort size from 25–3,000 trios, randomly generating variants and assigning to risk genes in agreement with the proportions seen in our data, then applying the TADA-Denovo algorithm. Based on these

simulations, WES of 500 trios should find 16 probable and 7 high-confidence risk genes; 1000 trios should find 51 probable and 26 high-confidence risk genes.
(TIF)

**S4 Fig. KDM5B brain expression levels.** Brain expression by region for KDM5B. Data is from GTEx Analysis Release V7 (dbGaP Accession phs000424.v7.p2) (https://gtexportal.org/home/gene/KDM5B). Expression values are shown in Transcripts Per Million (TPM), calculated from a gene model with isoforms collapsed to a single gene. No other normalization steps have been applied. Box plots are shown as median, 25th, and 75th percentiles. Points are displayed as outliers if they are above or below 1.5 times the interquartile range. Further details about expression quantification and samples can be found at https://gtexportal.org/home/documentationPage#AboutData.
(TIF)

**S5 Fig. KDM5B spatiotemporal brain expression.** Brain expression trajectories of KDM5B in the developing human brain. Expression data is from the Brainspan Consortium (brainspan.org, hbatlas.org), generated using the Affymetrix GeneChip Human Exon 1.0 ST Array platform. Vertical axis is the log2-transformed array signal intensity, which is proportional to transcript expression. A stringent threshold of $\geq 6$ was required to meet criteria for brain expression. Horizontal axis represents periods of human development and adulthood as previously defined by Kang et al (2011). Birth begins period 8 and adolescence begins period 12. Brain regions are by color: neocortex (NCX), hippocampus (HIP), amygdala (AMY), striatum (STR), mediodorsal nucleus of the thalamus (MD), cerebellar cortex (CBC).
(TIF)

**S6 Fig. Stereotypy scores in Simons Simplex Collection ASD probands.** Tukey box and whisker plot of Stereotyped Behavior Score (SBS) from the RBS-R (Repetitive Behavior Scale-Revised) in aged-matched Simons Simplex Collection ASD probands with (+, n = 4) and without (-, n = 364) de novo damaging mutations in KDM5B. Two-tailed Mann-Whitney test of ages (months) between groups: p = 0.86. One-tailed Mann-Whitney test of SBS between groups: p = 0.076.
(TIF)

**S1 Table. Phenotype, exome sequencing metrics, and principal components analysis.** See "S1 Table". First tab contains individual-level sample information (columns A-I), including family ID, individual ID, phenotype, cohort, collection site, gender, capture platform, size of "callable exome", and paternal age (years) at birth, where available. Column J lists reasons for any sample exclusions by quality control methods; "0" indicates that the sample was not excluded and was included in subsequent analyses. Columns K-AF list individual sample sequencing metrics generated using PicardTools, and GATK DepthOfCoverage tools. Columns AG-AQ list individual sample sequencing metrics generated using PLINK/SEQ (i-stats; https://psychgen.u.hpc.mssm.edu/plinkseq/stats.shtml). Columns B, K-AQ were included in Principal Components Analysis (PCA). Third tab contains cohort-level metrics calculated using samples passing quality control. ±95% confidence intervals are given, when applicable. Fourth tab contains coordinates generated for each sample for the top 10 principal components following PCA. The code used to generate this data is included in S1 File. Using these coordinates, we removed trios with family members falling more than three standard deviations from the mean in any of the first five principal components; this information is contained in the fifth tab.
(XLSX)

**S2 Table. Annotated de novo variants in pCMS and controls.** See "S2 Table". Detailed information on all high confidence de novo variants in cases and controls. These variants were annotated using ANNOVAR, based on RefSeq hg19 gene definitions. Column descriptions are provided in a separate tab of this file.
(XLSX)

**S3 Table. Gene-level de novo mutation rates, variant counts, and TADA-Denovo results.** See "S3 Table". First tab contains de novo mutation rates used to perform subsequent maximum likelihood estimation (MLE) and TADA-Denovo analyses. The following mutation rates are listed for each gene: overall (mut.rate), likely gene disrupting (lgd), predicted damaging missense (mis3, also referred to as Mis-D), and all damaging (lgd + mis3). These overall mutation rates were previously published (Ware et al., 2015) from unaffected parent-child trios. The code used to generate the mutation rate table is provided in S1 File. Second tab contains the input file for the TADA-Denovo algorithm. Gene-level expected mutation rates for LGD ("mut.cls1" column) and Mis-D variants ("mut.cls2" column) are listed, along with their respective observed mutation counts in our pCMS data ("dn.cls1" and "dn.cls2", respectively). Code for running TADA-Denovo is given in S1 File. Third tab contains the final output results from TADA-Denovo code provided in S1 File. One gene harboring more than one damaging de novo (LGD or Mis-D) variant in unrelated pCMS families is highlighted in yellow (KDM5B). This gene exceeded the threshold for being considered a high confidence (qval < 0.1) risk gene.
(XLSX)

**S4 Table. DNENRICH gene lists and results.** See "S4 Table". See S1 File for details of DNEN-RICH analysis and gene lists used. First tab contains input gene lists and information about their curation. Second tab contains the input mutation list for DNENRICH; each row represents a de novo damaging mutation in a pCMS proband. Third tab contains results output from DNENRICH. Significantly enriched gene sets are highlighted.
(XLSX)

**S5 Table. Exploratory pathway, gene ontology, and spatiotemporal analyses results.** See "S5 Table". Gene ontology results from Enrichr are in the first tab, including unadjusted and adjusted p-values. Gene ontology results from ConsensusPathDB are in the second tab; p-values < 0.05 and corresponding q-values are shown. Network analysis results from IPA are shown in the third tab. Specific Enrichment Analysis (SEA) exploring whether genes cluster within certain brain regions across development using Brainspan atlas data is in the fourth tab; p-values < 0.05 are highlighted in yellow. See S1 File for details of these analyses.
(XLSX)

**S1 File. Supplementary methods.**
(DOCX)

## Acknowledgments

We wish to thank the families who have participated in and contributed to this study. Control subject data were obtained from the NIH-supported National Database for Autism Research (NDAR). NDAR is a collaborative informatics system created by the National Institutes of Health to provide a national resource to support and accelerate research in autism. Dataset identifier: 2042. This manuscript reflects the views of the authors and may not reflect the opinions or views of the NIH or of the Submitters submitting original data to NDAR. We are grateful to all of the families at the participating Simons Simplex Collection (SSC) sites, as well as

the principal investigators (A. Beaudet, R. Bernier, J. Constantino, E. Cook, E. Fombonne, D. Geschwind, R. Goin-Kochel, E. Hanson, D. Grice, A. Klin, D. Ledbetter, C. Lord, C. Martin, D. Martin, R. Maxim, J. Miles, O. Ousley, K. Pelphrey, B. Peterson, J. Piggot, C. Saulnier, M. State, W. Stone, J. Sutcliffe, C. Walsh, Z. Warren, E. Wijsman). We appreciate obtaining access to phenotype and genetic data on SFARI Base. Approved researchers can obtain the SSC population dataset described in this study (https://www.sfari.org/resource/simons-simplex-collection/) by applying at https://base.sfari.org.

## Author Contributions

**Conceptualization:** Thomas V. Fernandez, Ellen J. Hoffman, Abha R. Gupta, Harvey S. Singer.

**Data curation:** Thomas V. Fernandez, Zsanett P. Williams, Tina Kline, Shreenath Rajendran, Farhan Augustine, Nicole Wright, Catherine A. W. Sullivan, Wenzhong Liu, Abha R. Gupta, Harvey S. Singer.

**Formal analysis:** Thomas V. Fernandez, Zsanett P. Williams, Abha R. Gupta.

**Funding acquisition:** Thomas V. Fernandez, Harvey S. Singer.

**Investigation:** Thomas V. Fernandez, Tina Kline, Farhan Augustine, Nicole Wright, Catherine A. W. Sullivan, Emily Olfson, Sarah B. Abdallah, Wenzhong Liu, Ellen J. Hoffman, Abha R. Gupta, Harvey S. Singer.

**Methodology:** Thomas V. Fernandez, Tina Kline, Shreenath Rajendran, Farhan Augustine, Nicole Wright, Catherine A. W. Sullivan, Emily Olfson, Sarah B. Abdallah, Wenzhong Liu, Ellen J. Hoffman, Harvey S. Singer.

**Project administration:** Thomas V. Fernandez, Zsanett P. Williams, Tina Kline, Shreenath Rajendran, Farhan Augustine, Nicole Wright, Wenzhong Liu, Harvey S. Singer.

**Resources:** Thomas V. Fernandez, Harvey S. Singer.

**Software:** Thomas V. Fernandez.

**Supervision:** Thomas V. Fernandez, Harvey S. Singer.

**Validation:** Thomas V. Fernandez, Shreenath Rajendran, Farhan Augustine, Nicole Wright, Catherine A. W. Sullivan, Wenzhong Liu, Abha R. Gupta, Harvey S. Singer.

**Visualization:** Thomas V. Fernandez.

**Writing – original draft:** Thomas V. Fernandez, Abha R. Gupta, Harvey S. Singer.

**Writing – review & editing:** Thomas V. Fernandez, Zsanett P. Williams, Tina Kline, Shreenath Rajendran, Nicole Wright, Catherine A. W. Sullivan, Emily Olfson, Sarah B. Abdallah, Wenzhong Liu, Ellen J. Hoffman, Abha R. Gupta, Harvey S. Singer.

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
