## [Decision Letter · Decision Letter 0]

18 Jun 2023

PONE-D-23-11138Primary complex motor stereotypies are associated with de novo damaging DNA coding mutations that identify KDM5B as a risk genePLOS ONE

Dear Dr. Thomas V. Fernandez

Thank you for submitting your manuscript to PLOS ONE. After careful consideration, we feel that it has merit but does not fully meet PLOS ONE’s publication criteria as it currently stands. Therefore, we invite you to submit a revised version of the manuscript that addresses the points raised during the review process.

We look forward to receiving your revised manuscript.

Kind regards,

Claudia Brogna

Academic Editor

PLOS ONE

Journal Requirements:

"This work was supported by grants from the Simons Foundation (sfari.org, SFARI award #239013, TVF), the Allison Family Foundation (TVF), Nesbitt-McMaster Foundation (HSS), Klump Family (HSS), and Graves Family (HSS)."

"I have read the journal's policy, and the authors of this manuscript have the following competing interests: Dr. Fernandez receives research/grant support from the National Institutes of Mental Health. Dr. Olfson receives research support from the National Institutes of Mental Health, the Alan B. Slifka Foundation through the Riva Ariella Ritvo endowment, and the International Obsessive-Compulsive Disorder Foundation. Dr. Singer serves as a consultant for Abide Therapeutics, Inc; Cello Health BioConsulting; ClearView Healthcare Partners; Teva Pharmaceutical Industries Ltd; and Trinity Partners, LLC. Dr. Singer receives publishing royalties from Elsevier and research/grant support from the Tourette Association of America. Other authors declare no potential conflicts."

Reviewers' comments:

Reviewer's Responses to Questions

**Comments to the Author**

1. Is the manuscript technically sound, and do the data support the conclusions?

Reviewer #1: Yes

Reviewer #2: Partly

2. Has the statistical analysis been performed appropriately and rigorously? 

Reviewer #1: Yes

Reviewer #2: I Don't Know

3. Have the authors made all data underlying the findings in their manuscript fully available?

Reviewer #1: Yes

Reviewer #2: Yes

4. Is the manuscript presented in an intelligible fashion and written in standard English?

Reviewer #1: Yes

Reviewer #2: Yes

5. Review Comments to the Author

Reviewer #1: This is a nice study on primary stereotypies that used WES and a de novo approach. The authors find an increased rate of damaging mutations in patients, and identify one likely candidate gene. Although it is affected by just 2 mutations, they are both stop/gain and I agree that this makes the gene a likely candidate. The authors them make several estimation regarding the likely number of genes involved and various gene-set analyses.

The only potential problem with the conclusions is the fact that the controls were genotyped separately as part of another study. The authors could add a paragraph explaining why this is not a problem and perhaps adding some data on sequencing depth and other features that could be different between the methods. But it is standard to combine large datasets of sequenced sampled from different labs, so I presume this was all done correctly.

Reviewer #2: In this manuscript, the authors used exome sequencing in parent-child trios and comparisons to a control cohort to identify genes and genetic mechanisms underlying primary motor stereotypes (pCMS). They report that relative to controls, pCMS cases show a modestly increased burden of damaging and likely damaging de novo genetic variants, a finding also described for other neuropsychiatric conditions. Two among the 118 pCMS cases in their cohort harbored two different stopgain de novo variants in KDM5B which was considered to exceed prespecified FDR significance criteria to nominate KDM5B as a pCMS risk gene. The authors further extrapolated from de novo damaging variant counts that a total of 184 genes might explain genetic liability to pCMS. Gene set enrichment suggest that the 52 genes carrying de novo damaging variants in pCMS cases are enriched for genes with described roles in ASD and Tourette syndrome as well as pCMS-related biological functions. This is a revised version of the manuscript, and the authors had already responded to comments from two reviewers in a previous round of review.

My main additional comment is that the authors’ response to Reviewer 1 does no longer hold up. Specifically, a recent paper by Chen et al. (Nature Genetics 2023 https://doi.org/10.1038/s41588-023-01398-8) establishes KDM5B as a dosage-sensitive gene for which heterozygote carriers show attenuated symptoms of phenotypes observed in individuals with autosomal-recessive KDM5B intellectual disability (ID) syndrome. Additionally, a more recent paper by Huang et al. (NCOMMS 2023; https://www.nature.com/articles/s41467-023-39247-1) introduce heterozygote variants in KDM5B as relevant for hand-grip strength. Importantly, Chen et al. also report that KDM5B has a fairly high de novo mutation rate, which sets it apart from the majority of other genes linked to ASD and ID. Notably, unlike the authors’ paper Chen et al. did not link KDM5B to pCMS (which does not seem to be well-curated in UK Biobank). They also followed a population rather than a trio-sequencing approach, which only indirectly allows for assumptions on whether variants are truly de novo. From this perspective, the current paper would certainly add to the literature if the authors assumptions for TADA-Denovo analyses (which I am not familiar with) and significance cut-offs for KDM5B remain unchanged in the light of variant frequency and pLi information from UK Biobank exome sequencing results.

6. PLOS authors have the option to publish the peer review history of their article (what does this mean?). If published, this will include your full peer review and any attached files.

Reviewer #1: No

Reviewer #2: No

---

## [Author Response · Author response to Decision Letter 0]

2 Aug 2023

Please see Response to Reviewers letter.

---

## [Decision Letter · Decision Letter 1]

10 Sep 2023

Primary complex motor stereotypies are associated with de novo damaging DNA coding mutations that identify KDM5B as a risk gene

PONE-D-23-11138R1

Dear Dr. Thomas V. Fernandez

We’re pleased to inform you that your manuscript has been judged scientifically suitable for publication and will be formally accepted for publication once it meets all outstanding technical requirements.

Kind regards,

Claudia Brogna

Academic Editor

PLOS ONE

Reviewer's Responses to Questions

**Comments to the Author**

1. If the authors have adequately addressed your comments raised in a previous round of review and you feel that this manuscript is now acceptable for publication, you may indicate that here to bypass the “Comments to the Author” section, enter your conflict of interest statement in the “Confidential to Editor” section, and submit your "Accept" recommendation.

Reviewer #2: All comments have been addressed

2. Is the manuscript technically sound, and do the data support the conclusions?

Reviewer #2: Yes

3. Has the statistical analysis been performed appropriately and rigorously? 

Reviewer #2: I Don't Know

4. Have the authors made all data underlying the findings in their manuscript fully available?

Reviewer #2: Yes

5. Is the manuscript presented in an intelligible fashion and written in standard English?

Reviewer #2: Yes

6. Review Comments to the Author

Reviewer #2: The authors now provide updated references and confirm that the newly emerged data do not change statistical conclusions of their original results. I do not have additional comments.

7. PLOS authors have the option to publish the peer review history of their article (what does this mean?). If published, this will include your full peer review and any attached files.

Reviewer #2: No

---

## [Editor Report · Acceptance letter]

25 Sep 2023

PONE-D-23-11138R1 

Primary complex motor stereotypies are associated with de novo damaging DNA coding mutations that identify *KDM5B* as a risk gene 

Dear Dr. Fernandez:

I'm pleased to inform you that your manuscript has been deemed suitable for publication in PLOS ONE. Congratulations! Your manuscript is now with our production department. 

Kind regards, 

on behalf of

Dr. Claudia Brogna 

Academic Editor

PLOS ONE